# Optimizing CO$_2$ Monitoring: Evaluating a Sensor Network Design

Kenia Elizabeth Sabando-Bravo [1], Marlon Navia [1,*] and Jorge Luis Zambrano-Martinez [1,2]

1   Departmento de Posgrado, Universidad Técnica de Manabí, Av. Urbina, Portoviejo 130104, Ecuador; keniasabando@gmail.com (K.E.S.-B.); jorge.zambrano01@utm.edu.ec or jorge.zambrano@uazuay.edu.ec (J.L.Z.-M.)
2   Computer Science Research and Development Laboratory, Universidad del Azuay, Cuenca 010204, Ecuador
*   Correspondence: marlon.navia@utm.edu.ec

## Abstract

In the present work, a sensor network design for monitoring carbon dioxide (CO$_2$) pollution in Portoviejo City, Ecuador, is evaluated through a methodology that combines simulation and physical implementation. This methodology involves the development and evaluation of two scenarios: an initial scenario (A), developed through both physical implementation and simulation, and another simulation scenario (B). Both simulated scenarios are created using CupCarbon version 6.51 software. In these scenarios, the functionality of Wireless Sensor Networks (WSNs) is analyzed by implementing the LoRaWAN communication technology. Furthermore, the MQ-135 sensor is used to obtaining data on the PPM of (CO$_2$) in order to examine the areas that concentrate the most significant amount of this atmospheric pollutant. The proposed networks are evaluated using the packet loss metric during data transmission. After implementation, analysis, and respective evaluation, it can be concluded that the network simulated in Scenario B is suitable for monitoring (CO$_2$) and other pollutants that can be analyzed within the urban environment.

**Keywords:** WSN; LoRaWAN; CO$_2$; CupCarbon; MQ-135



## 1. Introduction

Today, technology has advanced significantly in several areas and is being utilized to solve various problems, among them environmental pollution. Research is required in order to understand and solve issues related to air quality and its affects on people's health. One of the leading contributors to environmental pollution is Carbon Dioxide (CO$_2$). This gas contributes to the greenhouse effect by absorbing infrared radiation, resulting in an abnormal increase in surface temperature. For this reason, this pollutant has a greater influence on global warming than all gases combined [1].

Among the solutions technology currently offers for monitoring and controlling various gases are sensor networks. This type of technology comprises several sensors that allow for the measurement of atmospheric pollutants. It is very useful for detecting anomalies, allowing investigations of atmospheric behavior to be performed.

In this approach, devices known as sensor nodes are located at short distances. Their function is to collect various data about the environment through different sensors such as humidity, air quality, noise levels, movement, and temperature, among others. These networks can be applied in various areas or situations, such as health, safety, and agriculture [2].

Small sensors can be installed in buildings, factories, hospitals, schools, and other locations where accurate measurement of polluting gas levels such as $CO_2$, is required. This guarantees air quality and reduces respiratory problems. By grouping several sensors, a network can be configured to send real-time data of measurements via wireless communication.

Consequently, when $CO_2$ levels exceed certain thresholds, the sensor nodes autonomously trigger an alert, allowing administrators to take immediate action and improve air quality. Additionally, the data collected by the sensor network can be used to analyze trends and inform decisions about air quality in their respective locations [3]. $CO_2$ monitoring is essential for various industries where measuring indoor air quality and safety is vital for both industrial operations and scientific research. In indoor air quality, $CO_2$ monitoring determines whether levels of this gas are safe for human health and controls ventilation in buildings and closed spaces. In the security industry, it is used to detect gas leaks and ensure the safety of workers in hazardous areas. In scientific research, it is used to study the effects of climate change.

The objective of this research is to evaluate the design of a sensor network that enables the monitoring of $CO_2$ in Portoviejo City by combining simulation and prototype implementation of the design. In addition to the designed network, one of the main contribution of this paper lies in the methodology applied during the design and evaluation process of the Wireless Sensor Network (WSN). The methodology proposed in this research can be used by private companies, professionals, and research centers. In addition, the evaluated design can serve as a starting point for public institutions or governments, providing a basis for solutions to the excessive emission of this atmospheric pollutant, thereby improving urban territorial reorganization. The contributions are summarized below:

- Design and evaluation of a WSN for $CO_2$ monitoring in a specific city.
- A methodology for evaluating WSN design which combines simulation and real deployment.
- Preliminary measurements of $CO_2$ quantity at selected points of the city.

The rest of this paper is organized as follows: Section 2 presents a brief state-of-the-art review of $CO_2$ monitoring as well as other WSN applications; Section 3 describes the methodology applied in this work; Section 4 presents the results of the study, while Section 5 analyzes these results; finally, Section 6 presents the paper's conclusions.

## 2. Related Works

From experimental analysis to finding solutions to monitor air pollutants, several researchers have studied implementing sensor networks or Wireless Sensor Networks (WSNs) to monitor $CO_2$, mainly in outdoor environments.

Examples include the research by Bravo Granda et al. [4], where a sensor network was implemented in Cuenca, Ecuador. The MG-811 sensor was used to detect $CO_2$ levels at various points in the city. The results obtained by the authors were optimal, allowing essential conclusions to be drawn for decision-making on improving air quality.

In the Philippines, Palconit and Nuñez [5] developed a system for monitoring air pollutants through mobile sensor networks to address high $CO_2$ emissions from public transport under poor conditions. These nodes were installed in the exhaust pipes of public transport vehicles to collect data. The CO sensor they used was the NDIR COZIRWR-GC-0006, and they concluded that the highest emission of atmospheric pollutants occurs when public vehicles are stopped or moving uphill.

Implementing a sensor network can be very useful in situations such as natural disasters. In [6], a WSN was developed and implemented to monitor fires through the MG-811 sensor, allowing for the detection of $CO_2$ levels to identify disasters in the town.

To convert the city of Makassar, Indonesia into a smart city, Lisangan and Sumatra [6] implemented a sensor network using public infrastructure and the MQ-9 sensor to detect the presence of $CO_2$. Their objective was to utilize WiFi Corner as the infrastructure to detect congestion locations for the city government. They concluded that WSNs can be a valuable tool to achieve this objective [6].

In Taiwan, a water and air monitoring system was developed based on a sensor network utilizing Long Range Wide Area Network (LoRaWAN) communication, leveraging its advantages of low power consumption, energy efficiency, and long-distance capabilities. The system measured several parameters, focused on a campus area [7]. Another study by Montoya and Chilo [8] measured different atmospheric gases and communicated with a network of sensors in Juliaca, Peru. For this study, the MQ-135 sensor was used to monitor $CO_2$ gas, enabling the collection of necessary data to find solutions to the problem of atmospheric pollution in that area.

In Spain, it was proposed that a sensor network be implemented to control $CO_2$ emissions in the environment and make the necessary decisions to reduce pollution. For this study, the MQ-135 sensor was used, and it was concluded that LoRaWAN was an appropriate communication technology due to its scope for data collection in a city [9].

In 2020, Ordoñez Mendieta et al. [10] developed a sensor network implemented in Loja, Ecuador, intending to monitor Carbon Monoxide (CO) emissions and noise levels at specific city sites. They used CCS811 sensors placed in strategic areas for data collection to detect atmospheric pollutant levels.

Kitazumi et al. [11] proposed a WSN to measure $CO_2$ concentration, and studied its correlation with other environmental factors such as relative humidity at a university campus in Japan. They used LoRaWAN technology for communication, but the type or model of $CO_2$ sensor was not specified. In another study [12], a system with autonomous sensor nodes was developed for monitoring $CO_2$ in a Brazilian city with a high ratio of cars per inhabitant. The sensor nodes were made up of several components, including a Raspberry Pi, which increases cost but also provides greater autonomy.

Although the use of single modules or individual nodes to measure $CO_2$ is common, some proposals have combined multiple measurements to achieve more accurate values. In [13], a fusion estimation model based on the Kalman-Consensus filter was used to estimate the $CO_2$ concentration in a greenhouse. For this model, the WSN nodes are grouped in clusters to enhance the estimation.

Another study on data fusion was presented by [14], who proposed two sequential algorithms to perform a binary decision (presence or absence of a contaminant gas) by sequential detection, after which the individual decisions are transmitted to a fusion center. Despite the significant improvement in terms of detection accuracy and delay, the proposal does not show the quantity of gas detected.

Not all studies of $CO_2$ monitoring in air focus on outdoor environments; many are oriented toward indoor air monitoring and analysis. In [15], a WSN with low-power nodes was proposed to monitor indoor air quality. The researchers emphasized the importance of air monitoring in assessing its impact on human health [16].

The use of WSNs for $CO_2$ monitoring also extends to the soil. Hassan et al. [17] developed a probe system to monitor this substance in the soil, using LoRaWAN for communication between the probe nodes.

Simulators are essential in various scientific fields for determining behaviors in different circumstances. In [18], the authors studied a WSN to maximize the operation of a sensor network through the solar energy harvesting technique. Prior to implementing the sensor network, a simulation was performed in the Network Simulator (NetSim) program

using twenty nodes with an energy harvesting function and others without such a function. The use of solar energy harvesting extended the useful life of the WSN by 25%.

Saeed et al. [19] developed a study in which a fire alarm system was implemented through a wireless sensor network to facilitate early detection, thereby reducing damage and saving lives. This research utilized the Zigbee protoco. The proposal's functionality was evaluated using Fire Dynamic Simulator (FDS) simulation, which enables the creation of fire scenarios in various environments.

In [20], an evaluation of several routing protocols for a WSN is performed using the CupCarbon simulator. According to the authors, this simulator enabled the assessment of key metrics in a LoRaWAN network, including data package delivery rate, end-to-end delay, jitter, throughput, and more. In addition, CupCarbon can simulate not only routing protocols and power consumption but also advanced features such as encryption and authentication protocols [21].

LoRaWAN technology is widely used in monitoring proposals with both fixed and mobile sensor nodes. In [22,23], two different proposals for air quality monitoring using drones as sensor nodes were presented. These studies showed LoRaWAN to be a perfect choice for long-term tracking thanks to features such as low power consumption and long-range coverage.

Table 1 shows a categorization of the related works analyzed in this section. In terms of communication coverage, the distribution is similar among Wireless Wide-Area Network (WWAN), Wireless Local Area Network (WLAN), and Wireless Personal Area Network (WPAN) technologies; however, this is not necessarily the case for other criteria. Most researchers have proposed simple measurements as the data source, performed by sensor modules attached to nodes. As an alternative, data fusion can be performed on the measurements to achieve greater accuracy. For evaluation of monitoring proposals, the use of real hardware is the preferred method; though less widely applied, simulations can enable faster and easier evaluation of WSNs.

**Table 1.** Categorization of related works.

| Communication Technology | | | Measurement Mode | | Evaluation/Implementation | |
|---|---|---|---|---|---|---|
| WWAN (LoRa, 4G) | WLAN (WiFi, XBee) | WPAN (ZigBee) | Simple Sensors | Data Fusion | Real Hardware | Simulation |
| [7,9,11,12,22,23] | [5,6,8,16] | [4,10,15,18,19] | [4–12,15,16,18, 19,22,23] | [13,14] | [4–9,11,12,15,16, 22,23] | [10,13,14,18,19] |

## 3. Materials and Methods

To delineate the methodology employed in this study, an initial review was undertaken of various research works that have utilized sensor networks for the monitoring of multiple gases, including $CO_2$. The methodology in this paper aims to integrate the benefits of both simulation and real-world deployment. Accordingly, the methodology consists of the following phases:

- Collection and analysis of requirements
- Network design
- Design simulation
- Network implementation
- Data analysis.

### 3.1. Collection and Analysis of Requirements

We systematically reviewed tools and technologies relevant to WSNs for environmental monitoring in order to establish a robust methodological framework. This review

encompassed both specific case requirements and the components and technologies pertinent to WSN design.

Initially, fundamental requirements were delineated (e.g., $CO_2$ monitoring in Portoviejo City, employment of remote nodes, and conducting a realistic evaluation), followed by the selection of components for the WSN design. The subsequent subsections elaborate on the selection criteria and provide the justification for each component.

### 3.1.1. Communication Technology

Our research began with a practical approach by seeking communication technologies that could be feasibly implemented for our study. The study adopted a pragmatic approach to evaluating communication technologies suitable for large-scale urban deployment. After benchmarking alternatives (e.g., Zigbee, NB-IoT, and Sigfox), LoRaWAN was selected due to its features.

LoRaWAN is an Low-Power Wide Area Network (LPWAN) technology. It is an open-source technology focused on providing wide coverage with minimal resource requirements. LoRaWAN operates in open bands, which allowed us to develop a private network without requiring third-party infrastructure. Its long-range wireless capability, location, and outdoor implementation suitability make it a practical choice for research on energy consumption and large-area deployments [24–26]. As shown in Table 1, most of the proposals for $CO_2$ monitoring in the WWAN paradigm utilize LoRaWAN as a communications technology. In [27], a detailed description and analysis of this technology is performed. Among the main advantages of LoRaWAN are the following:

- Long-range capability (2–15 km in urban areas)
- Low energy consumption, enabling multi-year battery life for sensors
- Scalability, supporting thousands of nodes per gateway.

These attributes align with the requirements for monitoring $CO_2$ levels across a city, particularly in areas with limitations or restrictions.

### 3.1.2. $CO_2$ Sensor Selection

Our comparative analysis of $CO_2$ sensors focused on accuracy, cost, and compatibility with IoT networks. As detailed in [28], the MQ-135 semiconductor sensor was chosen due to its features:

- Sensitivity range (10–1000 ppm $CO_2$)
- Low power consumption (~150 mW) [29]
- Validation in peer-reviewed studies for urban air quality monitoring [30].

While electrochemical sensors (e.g., Sensirion SCD30) or Non-Dispersive Infrared (NDIR) sensors (e.g., MH-Z19) can offer higher precision [31], the MQ-135 sensor provides an optimal tradeoff between performance and budget constraints [32]. While this sensor is not considered a highly accurate sensor, it is useful for evaluating the designed WSN.

To calibrate the sensor measurements for $CO_2$, we adjusted the resistance of the sensor module according to the manufacturer's indications. To obtain the measurements in ppm (parts per million), we applied Equation (1), where $a$ and $b$ are constants that depend on the specific gas to be measured, $Ro$ is the sensor's resistance in clean air, and $Rs$ is the sensor's resistance at various concentrations of gases [33]. All of these values have previously been determined for $CO_2$. A limitation of this sensor is that the manufacturer does not indicate the error range, although it does mention that environmental conditions can affect the measurements.

$$ppm = a * (Rs/Ro)^{\wedge}b \tag{1}$$

### 3.1.3. WSN Simulator Tool

The simulator chosen for this research was CupCarbon version 6.51, which was selected because it allows a representation that is very close to that of a real network. CupCarbon can also perform analyses of different variables in a wireless network designed for IoT devices. Additionally, its work interface is visual and user-friendly, making the learning curve for designing a network relatively fast. Using this simulator, we leverage the following features:

- Realistic node mobility and energy consumption algorithms [34]
- Integration of LoRaWAN protocols [35]
- Ability to simulate large-scale IoT deployments with terrain-specific propagation models [36].

In our specific case, we wanted to evaluate CO2 monitoring in a city. The CupCarbon simulator enabled the use of LoRaWAN with rapid implementation capabilities to simulate faults and visualize locations on a real map.

### 3.2. Network Design

For the physical implementation and simulation of the sensor network, LoRaWAN communication technology was utilized because it possesses the necessary characteristics to transmit and receive data over long distances. For this study, the data generated by the sensors that monitor gases in a city are both sent and received. LoRaWAN was chosen for its range of up to 15 km in line of sight, low consumption, greater fault tolerance, and better signal propagation.

Our study concerns not just implementation but also a thorough evaluation; thus, we simulated two network scenarios for evaluation to determine which best suited our study's needs. We then implemented one of the simulated scenarios to test its functionality in the real world, ensuring a comprehensive evaluation process.

In the simulated Scenario A, we strategically placed three sensor nodes at various points in Portoviejo City. These nodes were not randomly placed; rather, they were intentionally located at significant distances to allow for a better network evaluation and to obtain $CO_2$ measurements at these points. The precise locations are detailed in Table 2.

**Table 2.** Location of sensor nodes in Scenario A.

| Node | Location | Geographic Coordinates |
|---|---|---|
| Node 1 | Chile St. and Ramos Iduarte Ave. | −1.0521625412469338, −80.45612958616223 |
| Node 2 | Pablo Zamora Ave. | −1.0428113257944391, −80.45904501135622 |
| Node 3 | UTM Human Resources Department | −1.0461644648065493, −80.45518665109073 |
| Gateway | UTM Faculty of Computer Science | −1.0411741196059903, −80.456772111608543 |

The locations of nodes in the simulated Scenario A were chosen based on observations of areas with a more significant presence of vehicles. Another reason for their location was accessibility for placement. For the physical implementation of the network, the characteristics and sensor locations of Scenario A were chosen; three physical nodes were created, each with a sensor for $CO_2$ monitoring. Node 1 was located in the Nilton Díaz Building, where the offices of the Provincial Government of Manabí currently operate; Node 2 was placed in a house on the side of Rotonda Park; Node 3 was placed outside the

Human Resources Department of the Universidad Técnica de Manabí (UTM); finally, the gateway was located in the Faculty of Computer Sciences of the UTM (Figure 1).

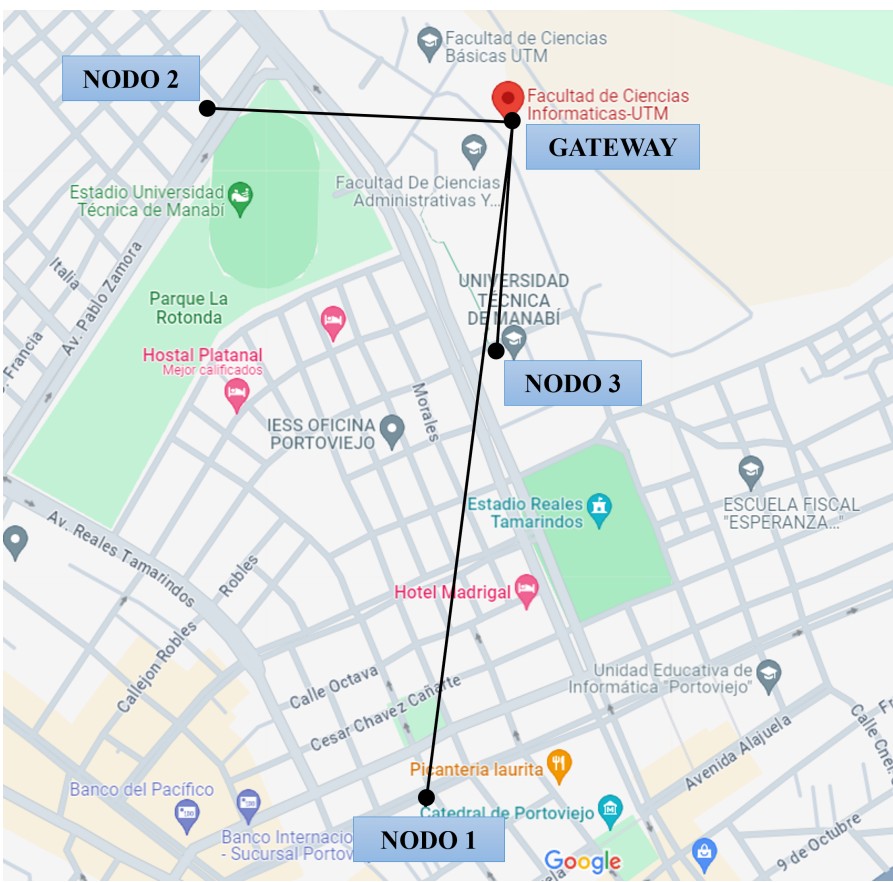

**Figure 1.** Scenario A physical design (node locations).

For the simulated Scenario B, a gateway and nine sensor nodes were distributed throughout the city at varying distances (Figure 2).

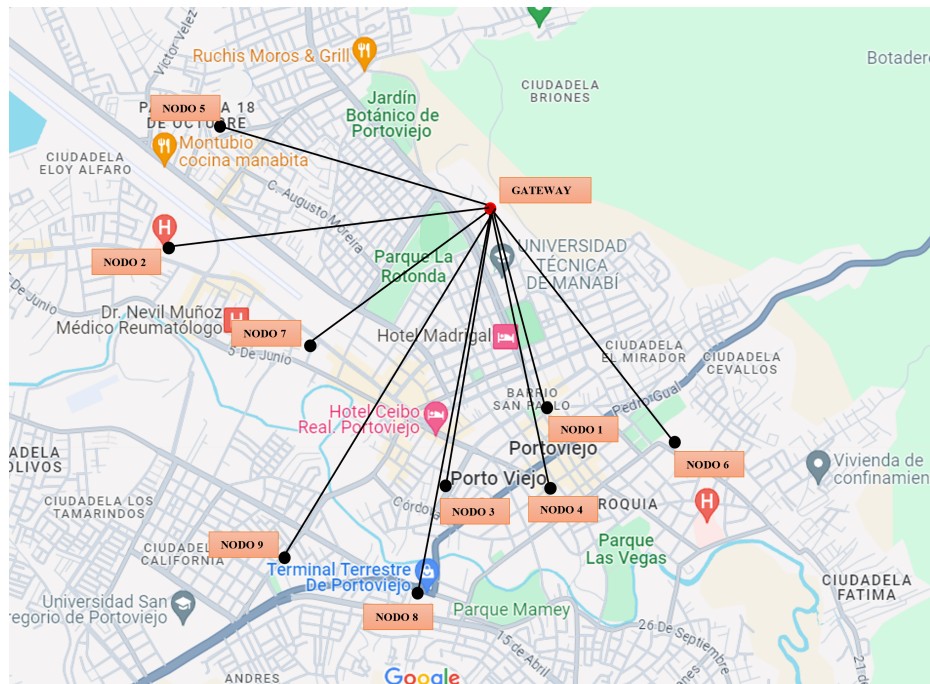

**Figure 2.** Scenario B physical design (node locations).

The locations for Scenario B are detailed in Table 3. These locations were selected after a thorough analysis because they have the highest vehicle traffic, vehicle congestion, levels of smoke, and other factors. This selection process ensures the effectiveness of the network implementation and accuracy of $CO_2$ monitoring.

**Table 3.** Location of sensor nodes in Scenario B.

| Node | Location | Geographic Coordinates |
|------|----------|------------------------|
| Nodo 1 | José María Urbina Ave. and Francisco de P Moreira St. | $-1.05253, -80.45278$ |
| Nodo 2 | Manabí Ave. | $-1.04472, -80.47185$ |
| Nodo 3 | Alajuela Ave. | $-1.05694, -80.45799$ |
| Nodo 4 | 10 de Agosto St. and Chile Ave. | $-1.05689, -80.45291$ |
| Nodo 5 | Eduardo Loor St. | $-1.03865, -80.46877$ |
| Nodo 6 | Cristo Rey St. | $-1.05384, -80.44579$ |
| Nodo 7 | 5 de Junio Ave. | $-1.05052, -80.46440$ |
| Nodo 8 | Army Ave. | $-1.06186, -80.45876$ |
| Nodo 9 | America Ave. and 5ta. Transversal St. | $-1.06051, -80.46552$ |
| Gateway | UTM Faculty of Computer Science | $-1.0411741196059903, -80.456772111608543$ |

The selection of locations for Scenario B aimed to consider areas with potentially high levels of contamination as well as to evaluate different distances and environmental conditions. On the other hand, locations for Scenario A were selected to evaluate connectivity and $CO_2$ hllevels as well as to provide an initial prototype of the monitoring WSN.

### 3.3. Simulation of the Network Design

The simulator used for the network designs was CupCarbon, since it allows for simulation through the Long Range (LoRa) communication module and for the reasons explained earlier.

The central research scenario was conducted in the selected city, with sensor nodes located in the respective geographic coordinates according to each simulated scenario. After the sensor nodes were placed in the simulation along with the respective programming for sending data, which included setting the data transmission frequency, packet size, and other parameters, the simulation automatically placed communication lines between all nodes. However, these are not connected, nor do they allow for jumps between the nodes; rather, an indication is made to signal the presence of the signal between the nodes.

At this stage, we executed simulated scenarios A and B to determine the percentage of packets sent from each node and the viability of these nodes. To obtain data closer to the real situation, transmission faults were included in each simulation (i.e., messages that are not received by the gateway) using an option provided by the simulator. The reason for this is that in a real environment, not all messages transmitted over the WSN will arrive at the gateway.

### 3.4. Implementation of the Network Design

For the physical implementation of the sensor network, we chose Scenario A. This decision was based on the economic cost involved in developing the nodes and the ease of access to their locations, which affected the practicality of our project. In this implementation, we deployed three sensor nodes, each equipped with the gateway and LoRa communication technology.

Furthermore, we implemented the MQ-135 sensor in each physical node for $CO_2$ detection. This sensor was chosen for its high sensitivity in capturing atmospheric pollutants,

which ensures accurate data, as well as for its low economic cost. Figure 3 provides a visual depiction of one of the nodes used in the implementation.

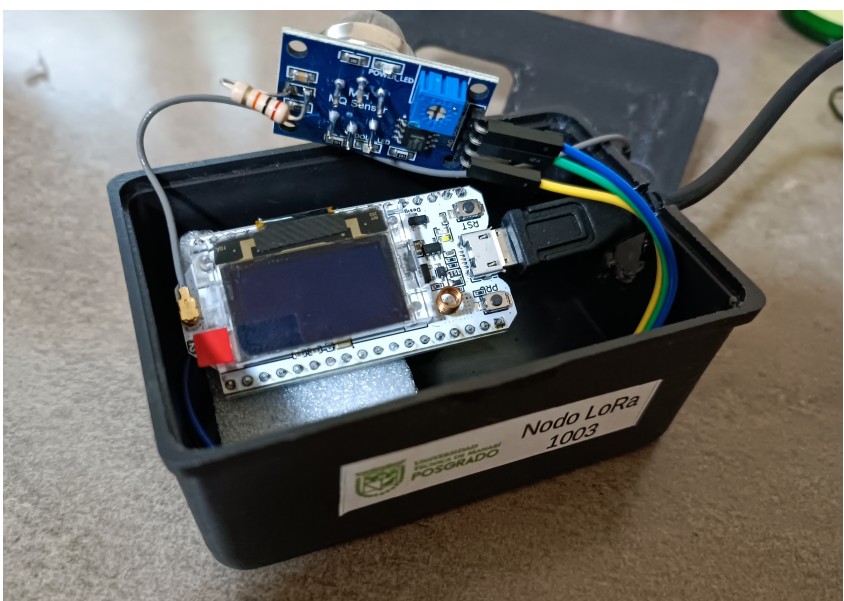

**Figure 3.** Sensor node used in physical implementation.

In addition, a gateway based on a Dragino LG01-N device (Figure 4) was used for data collection. A connection was configured to the ThingSpeak platform, allowing for visualization of data received every 30 min from sensor nodes located at the three points in the city.

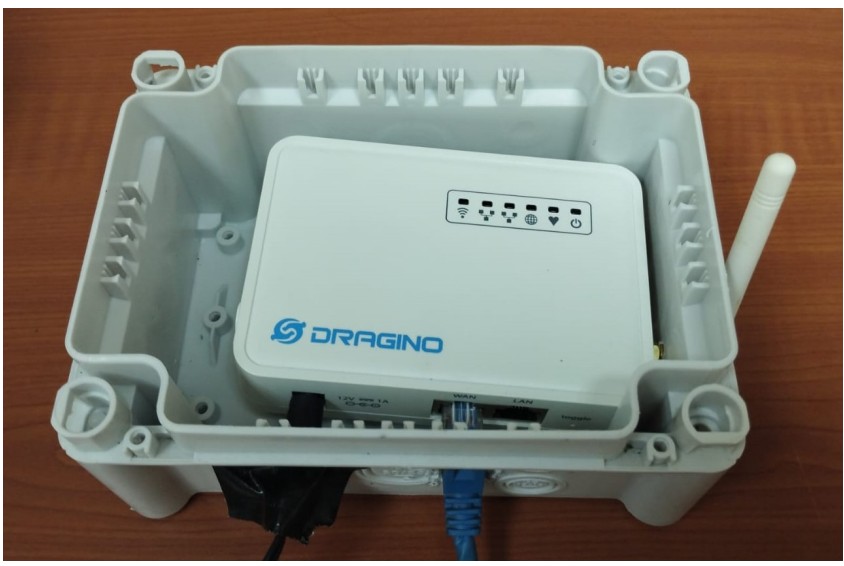

**Figure 4.** LoRaWAN gateway for the physical implementation.

Because the network uses LoRa communication technology, it was selected on the platform and the respective configuration was made to establish communication, where the server port, gateway ID, and other parameters were entered. Then, server configuration was performed by entering parameters such as the username, password, user ID, port for data transmission, and other connection parameters.

The respective configuration was performed for communication using LoRaWAN for each sensor node. Table 4 lists the configuration parameters assigned to each node. In this configuration, a Spreading Factor (SF) of 7 (i.e., SF7) was selected, with a bandwidth

of 125 MHz. The SF is a parameter ranging from 7 to 12 that controls the speed of data transmission and the sensitivity of the receiver, influencing both the range and power consumption. A higher SF means a higher data rate and lower area coverage; on the other hand, a lower SF indicates a lower data rate and higher area coverage. We chose the SF7 because the distance to the sensor nodes was not very great.

**Table 4.** LoRaWAN configuration of sensor nodes.

| Parameter | Configuration |
|---|---|
| Bandwidth | 125 KHz |
| Spread factor | SF7 |
| Length of preamble | 8 bits |
| Frequency | 915,600,000 Hz |

These parameters were programmed into the three physically implemented sensor nodes along with the server port, gateway ID, and other necessary details. This process was crucial because it allowed us to establish communication using the LoRaWAN protocol, ensuring the necessary transmission range between the nodes.

Each sensor node has its own respective Message Queuing Telemetry Transport (MQTT) channel for storing the information obtained through the $CO_2$ sensor. In the ThingSpeak platform, each channel displays the data stored by each node, with saved data downloaded in the form of an Excel file.

### 3.5. Data Analysis

After simulating the sensor network using CupCarbon and executing the physical implementation of the sensor nodes at various points in the city, we conducted a statistical analysis. This analysis was crucial because it allowed us to compare the data generated by the simulator with the data obtained from the physical nodes. We specifically examined the number of sent and lost packets, which provided valuable insights into the network's performance and reliability.

Additionally, the ThingSpeak Internet of Things (IoT) platform was utilized to store the data collected from the sensor nodes as well as to display and download it.

For data distribution analysis, the statistical measure called kurtosis was applied. This measure allowed us to determine the shape of the data distribution and identify whether the data distribution had heavy or light tails compared to a normal distribution. Then, the *t*-test for unequal variances was applied (Equation (2)), where the respective analysis sought to detect any significant difference between the means of the two designs.

$$t = \frac{\bar{x}_1 - \bar{x}_2}{\sqrt{\frac{s_1^2}{n_1} + \frac{s_2^2}{n_2}}} \tag{2}$$

## 4. Results and Validation

In this section, we present the results of the tests of both scenarios; the first scenario included both simulation and real deployment, while the second was purely a simulation. Both scenarios are of equal importance and provide valuable insights into packet loss in different evaluation methods.

### 4.1. Scenario A

In the first simulation scenario (comprising three nodes), we measured both the absolute quantity and the relative percentage of lost packets from a predefined number of transmitted packets. The simulation covered three days of operation, with each node

transmitting 240 packets per day, analyzed independently. To enhance the realism of the results, we deliberately introduced controlled fault injections during the simulation runtime. The aggregated results for the three-day period are presented in Figure 5, which summarizes the packet loss trends across all nodes.

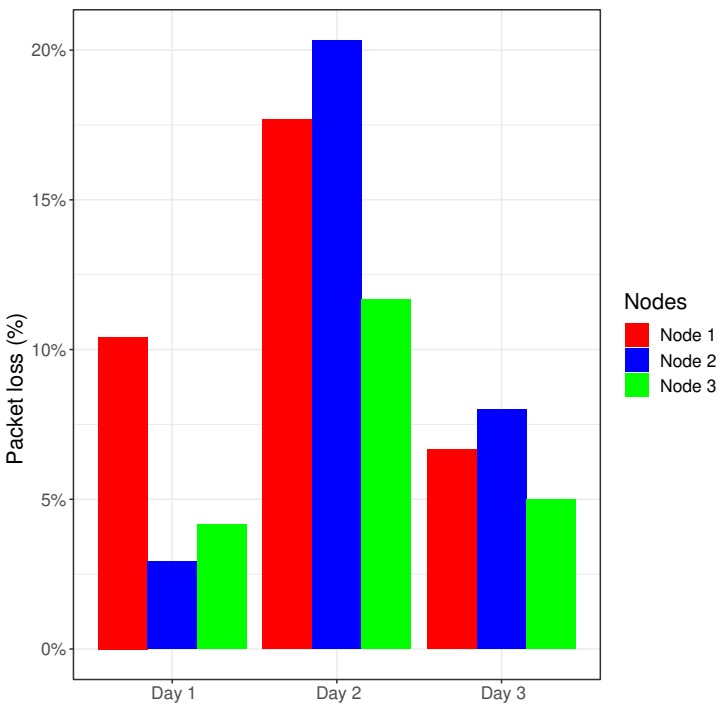

**Figure 5.** Percentage of packet loss over three days of network monitoring in simulation Scenario A.

As illustrated in Figure 5, the packet loss percentages varied significantly across nodes during the three-day simulation. On the first day, Node 1 exhibited the highest packet loss (10.42%), followed by Node 3 (4.17%) and Node 2 (2.92%). The trend shifted on the second day, with Node 2 experiencing the most substantial loss (20.33%), while Node 1 and Node 3 recorded losses of 17.67% and 11.67%, respectively. By the third day, Node 2 again led in packet loss (8%), though at a reduced rate compared to the previous day, while Node 1 and Node 3 showed further declines (6.67% and 5%, respectively). These fluctuations may be attributed to simulated network conditions, including variable node-to-gateway distances and dynamic fault injections designed to replicate real-world connectivity challenges.

In the real-world deployment, three sensor nodes were strategically positioned across the city to monitor $CO_2$ levels over a one-week period in July. Each node transmitted air pollutant data to the gateway at 30-minute intervals, enabling continuous environmental monitoring. This setup facilitated a robust evaluation of network reliability under operational conditions. A comparative analysis of simulation and physical deployment results provides critical insights into the system's performance, particularly in terms of packet loss dynamics. The daily packet loss percentages for each node in the physical network are detailed in Figure 6.

The second day of monitoring revealed notable packet loss patterns across the network nodes. Both Node 1 and Node 2 exhibited identical loss percentages of 16.67%, suggesting potential similarities in their network conditions or operational constraints. Node 3 demonstrated significantly better performance with only 8.33% packet loss, representing the most reliable communication link among all nodes during this observation period. The substantial difference in Node 3's performance may indicate either more favorable positioning, superior hardware resilience, or more stable connection pathways to the gateway.

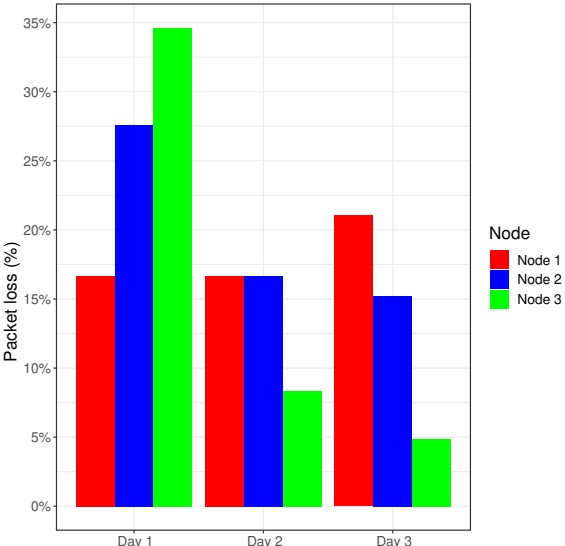

**Figure 6.** Percentage of packet loss over three days of network monitoring for the physical deployment.

A significant change in network behavior was observed on the third day, as illustrated in Figure 6. Node 1's packet loss increased to 21.05%, representing the highest value recorded across all measurement intervals, possibly indicating emerging connectivity difficulties or heightened network congestion. Node 2 maintained relatively stable performance with a loss rate of 15.22%, showing only minimal improvement from the previous day. Notably, Node 3 continued its exemplary performance trend, attaining the lowest loss rate of 4.88% within the network. This consistent superior performance by Node 3 across multiple measurement periods merits further examination of its technical specifications or topological advantages within the network architecture.

The analysis of packet loss percentages across the three sensor nodes revealed a consistent performance pattern, with Node 2 exhibiting the highest packet loss rates, followed closely by Node 1. In contrast, Node 3 showed significantly better performance. This performance degradation in Nodes 1 and 2 can be attributed to multiple compounding factors: (1) their greater physical distance from the Gateway, resulting in weaker signal strength; (2) substantial urban interference from tall buildings and other structures obstructing the line-of-sight communication paths; and (3) adverse weather conditions, particularly rainfall, which further attenuated signal quality through atmospheric absorption and scattering effects. Indeed, on the first day of the test, there was a slight rain in the city. Historical weather data, obtained from https://www.tutiempo.net/clima/ws-841350.html (accessed on 9 September 2025), confirm the following precipitation measurements of rain for the days of the tests:

- Day 1: 4.06 mm.
- Day 2: 2.03 mm.
- Day 3: 0.0 mm.

In addition, there was no clear line of sight between the nodes and the gateway. These environmental challenges collectively contributed to the observed disparity in network performance across nodes.

To evaluate the operational effectiveness of the deployed network architecture, environmental sensor data were systematically collected and logged via the Thinkspeak IoT platform. Figure 7 presents the temporal variation of atmospheric $CO_2$ concentrations (measured in ppm) across all monitoring locations during the 72-h observation period. Spatial

analysis reveals pronounced heterogeneity in pollution levels, with Node 2 consistently recording the highest $CO_2$ concentrations greater than other nodes.

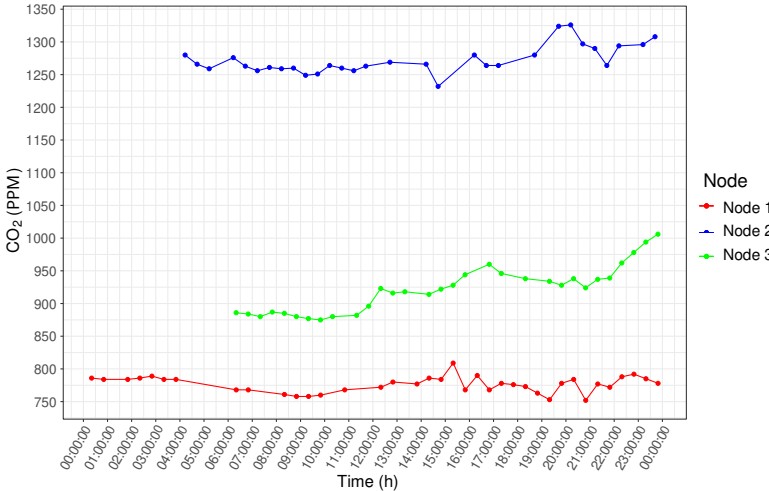

(**a**) $CO_2$ concentration (ppm) on day 1.

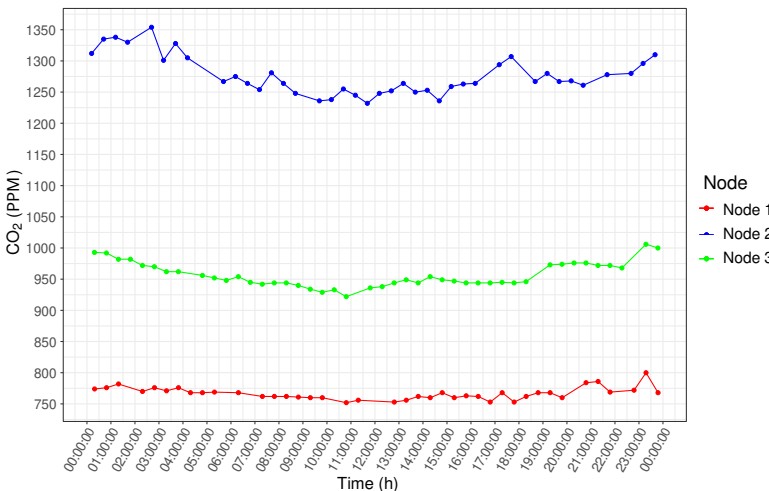

(**b**) $CO_2$ concentration (ppm) on day 2.

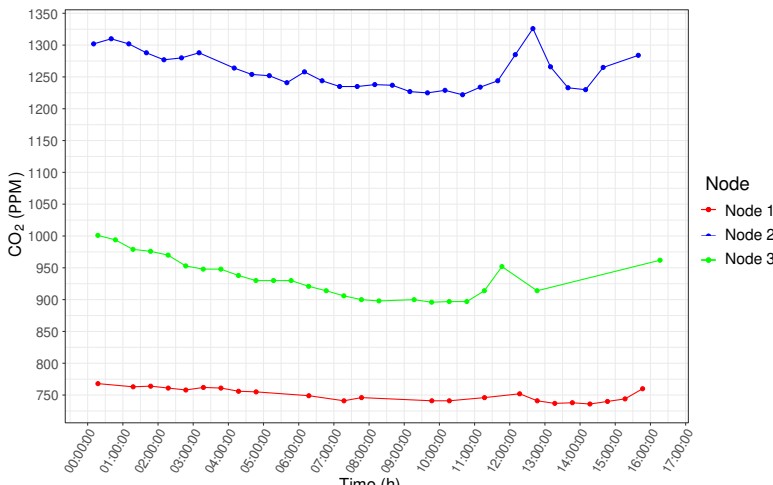

(**c**) $CO_2$ concentration (ppm) on day 3.

**Figure 7.** Evolution of $CO_2$ concentration (in parts per million) over three days of continuous monitoring in Scenario A.

Although the elevated pollution signature could correlate with the node's strategic placement in a high-traffic urban corridor characterized by frequent vehicular congestion where idling vehicles and stop-and-go traffic patterns are known to exacerbate localized emissions, the high values can also correspond to an unusual operation of the sensor due to its limitations. As such, the empirical data validate both the network's monitoring capabilities and the anticipated spatial distribution of urban air pollutants. The acquired data will be made available as part of this publication.

In Table 5, a descriptive analysis of measured $CO_2$ is presented. This analysis shows that the values obtained in Node 1 have fewer variations in its measurements. Although Nodes 2 and 3 have different means, their variability is highly similar.

**Table 5.** Descriptive comparison of $CO_2$ measurements.

| Parameters | Node 1 | Node 2 | Node 3 |
|---|---|---|---|
| Mean | 765.13 | 1271.59 | 885.85 |
| Variance | 197.06 | 1477.63 | 1454.37 |
| Standard deviation | 14.04 | 38.44 | 38.14 |
| Kurtosis | 0.446 | 11.41 | 0.50 |

*4.2. Scenario B*

The sensor network in Scenario B was systematically simulated in order to quantify packet loss percentages, facilitating a comprehensive evaluation of the network's reliability and performance. These results were subsequently compared with those obtained from Scenario A to assess relative performance under differing configurations. As illustrated in Figure 8, the simulation data disclose notable variations in packet loss across nodes, indicating differential influences of network topology or environmental conditions. This comparative analysis offers critical insights into the robustness of the network design under varying operational parameters.

The initial day of the simulation revealed significant disparities in node performance. Node 1 (4%) and Node 7 (4.33%) proved to be the most dependable, demonstrating minimal packet loss. Conversely, Nodes 3 (17.00%), 8 (16%), and 9 (15.33%) exhibited considerably reduced performance, suggesting the presence of potential bottlenecks or vulnerabilities within the network architecture. Intermediate nodes displayed packet loss rates ranging that these extremes, indicating a gradient of network reliability that may be associated with their physical placement or communication pathways. These preliminary findings underscore the need for targeted optimization of the less performant nodes.

A notable change in network performance was detected on the second day of simulation. Node 2 experienced a significant increase in packet loss, reaching 24%, which marked the most considerable degradation observed throughout the study period; conversely, Node 4 exhibited exceptional reliability, with a packet loss of only 5.67%. It is noteworthy that the majority of nodes (n = 7) showed enhanced stability, with loss rates remaining below 15%. This day-to-day fluctuation may indicate dynamic network conditions or the cumulative impact of simulated environmental stressors, necessitating further research into temporal performance variations.

The final day's simulation revealed a partial recovery of network performance, with Node 7 achieving the lowest loss rate (5%) and Node 3 persisting as the most problematic node (20% loss). The remaining nodes exhibited moderate losses clustered between 7% and 15.67%, suggesting the establishment of more stable communication pathways. When analyzed collectively, the three-day dataset demonstrates that while specific nodes (3 and 2) consistently underperformed, others (4 and 7) maintained robust connectivity, poten-

tially indicating optimal placement or superior hardware configurations. These findings underscore the importance of node-level diagnostics in network optimization strategies.

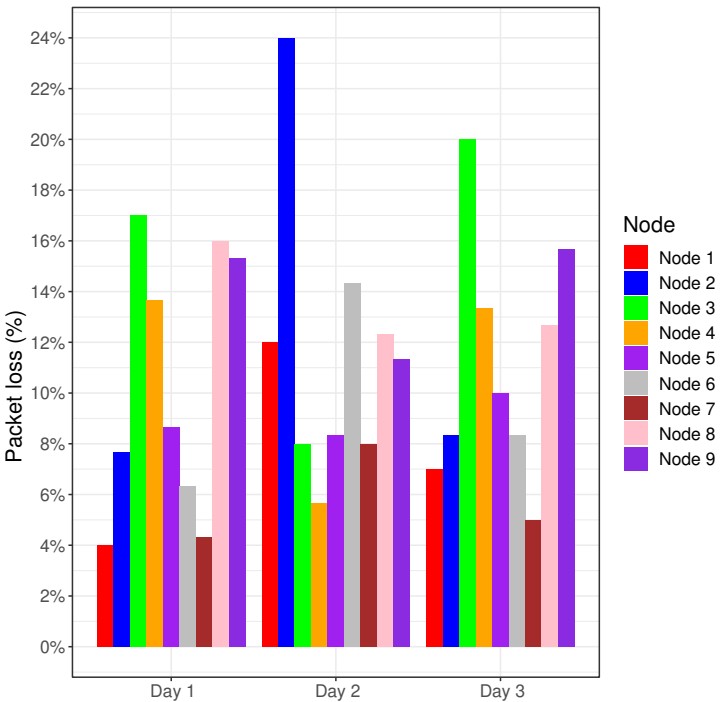

**Figure 8.** Percentage of packet loss over three days of network monitoring through simulation for Scenario B.

### 4.3. Comparative Analysis

Preliminary analysis of the WSN transmission test datasets indicated deviations from ideal normality; however, the concentration of values near zero implied an approximately normal distribution. Considering these distribution characteristics and the relatively small sample sizes in each case, Welch's *t*-test for unequal variances was employed to ensure a robust statistical evaluation. This methodology is particularly suitable for comparing datasets with potentially heteroskedastic variances, as it does not presuppose equal sample sizes or variances between groups.

The main aim of this statistical analysis was to determine whether the simulated Scenario A could validly approximate the behavior of the physical implementation under similar design conditions. A two-tailed *t*-test was performed to compare daily packet loss percentages across all nodes between the two scenarios, with the significance level ($\alpha$) established at 0.01 to increase the credibility of the results. This threshold was selected to reduce the probability of Type I errors while ensuring sufficient statistical power in light of our sample limitations.

As presented in Table 6, the analysis yielded a two-tailed *p*-value of 0.04781, which marginally exceeds our $\alpha$ value. Thus, under these conditions there is no evidence for rejecting the hypothesis that both tests are statistically similar. The *p*-value remains slightly below the conventional significance level of 0.05, which is used in most studies that apply this statistical test. This result provides moderate evidence supporting the hypothesis that the simulation scenario approximates the physical implementation behavior ($p = 0.048$).

Although the observed variance in the test data indicates some operational differences between simulation and physical implementation of Scenario A, the overall similarity in central tendency measures suggests the simulation's practical utility for network evaluation purposes. The moderate significance level ($0.01 < p < 0.05$) implies that while the tests are

not statistically identical at a stronger threshold, the simulation demonstrates sufficient fidelity for most engineering applications where complete correspondence is not required.

**Table 6.** Data obtained from the *t*-test between the simulated Scenario A and the physical deployment.

|  | **Physical Deployment** | **Simulation** |
| --- | --- | --- |
| Mean | 17.96 | 10.34 |
| Variance | 81.99 | 27.55 |
| Sample size | 9 | 9 |
| T value | 2.18 | - |
| P(T ≤ t) one tail | 0.02390 | - |
| P(T ≤ t) two tails | 0.04781 | - |

A possible limitation of this analysis is that the sample size may be small in the first analysis (implementation and simulation of Scenario A). In the same analysis, a value for $\alpha$ of 0.01 can be seen as non-significant.

An two-sample *t*-test was also conducted to compare the performance of the simulated scenarios (A and B), assessing whether the increase in the number of nodes led to a statistically significant difference in network performance. Again, the analysis was performed using a two-tailed test with a significance level ($\alpha$) of 0.01. The results summarized in Table 7 include key metrics such as the mean values, variance, sample sizes, t-statistic, and corresponding *p*-values.

**Table 7.** Data obtained from the *t*-test between the simulated Scenarios A and B.

|  | **Scenario A** | **Scenario B** |
| --- | --- | --- |
| Mean | 10.34 | 11.01 |
| Variance | 27.55 | 24.36 |
| Sample size | 9 | 27 |
| T value | −0.34 | - |
| P(T ≤ t) one tail | 0.37096 | - |
| P(T ≤ t) two tails | 0.74192 | - |

The two-tailed *p*-value of 0.74192 ($p > \alpha$) indicates that the difference in performance between Scenario A and Scenario B is not statistically significant at the confidence level of 1%. Although an $\alpha$ value of 0.05 can be used, the *p*-value remains lower than this value. This suggests that increasing the number of nodes did not substantially alter the network's performance under the tested conditions. Consequently, both configurations remain viable options for implementing a $CO_2$ monitoring network, with no significant advantage of one over the other in terms of the measured metrics.

## 5. Discussion

The comparative analysis between the simulations of Scenario A and Scenario B indicates an absence of statistically significant differences in performance across the two network architectures. Likewise, a comparison between the simulated outcomes of Scenario A and its physical implementation shows no substantial discrepancies. Nevertheless, marginally increased packet loss rates were observed during the real-world deployment, presumably due to environmental influences or signal interference.

From a practical deployment perspective, the higher node density in Scenario B could offer better spatial coverage for urban monitoring, as it allows for data collection from a larger number of points across the city. Nevertheless, Scenario A was selected for the initial implementation due to its lower resource requirements and ease of deployment, while still providing sufficient data to validate the network design.

The selection of LoRaWAN as the communication technology has been demonstrated to be highly appropriate for urban environments, corroborating previous research findings such as those presented by [9]. Additionally, air quality monitoring capabilities can be expanded beyond stationary sensor nodes through the integration of mobile sensing units, as evidenced in [37], where LoRaWAN was also employed for effective data transmission.

The real-world data collected from the deployment of Scenario A provided preliminary insights into $CO_2$ concentrations across selected urban areas. While broader spatial coverage would strengthen the dataset, the current findings still offer a valuable foundation for local policymakers to assess air quality trends. However, the cause of the unusual values obtained by one of the nodes must first be identified.

We have not found any proposal for $CO_2$ monitoring in this city in scientific literature that would allow for comparison with our design. However, comparison with similar proposals in the literature demonstrates our designed of a WSN with low-cost and long-distance communication nodes that can be used for monitoring not only $CO_2$ but also other gases of interest.

Although this study primarily concentrated on WSN design optimization for $CO_2$ monitoring, future developments in sensor node technology may encompass the integration of supplementary air quality sensors, such as those for particulate matter and Nitrogen dioxide ($NO_2$). This expansion, similar to the multi-sensor methodology described in [38], would enhance the versatility of the nodes while preserving the low-power and wide-area benefits inherent in the proposed design.

## 6. Conclusions

This study presents a comprehensive methodology for analyzing and designing a WSN for $CO_2$ monitoring, combining simulation-based evaluation with targeted real-world deployment to optimize network performance assessment while minimizing resource expenditure. By leveraging this dual-phase approach, we demonstrate that robust WSN validation can be achieved without full-scale physical implementation, offering a cost-effective framework for urban air quality monitoring systems.

$CO_2$ emissions, predominantly originating from transportation systems, constitute a critical environmental challenge in modern cities. To address this issue, we evaluated a WSN design for $CO_2$ monitoring through simulations and physical deployment in Portoviejo City, Ecuador, an urban environment facing growing air quality concerns. Our literature review confirmed that the MQ-135 sensor paired with LoRaWAN communication remains the prevalent choice for large-scale air quality monitoring due to its cost efficiency, reliability, and low energy consumption, attributes that were successfully replicated in our implementation.

Using the CupCarbon simulator, we modeled two distinct WSN scenarios (A and B) and physically deployed Scenario A across three strategic locations to validate simulation accuracy. However, our findings reveal that Scenario B presents the most viable solution for future deployments thanks to its enhanced node distribution, achieving optimal coverage while maintaining energy efficiency.

The real-world implementation of Scenario A not only confirmed CupCarbon's reliability as a simulation tool but also yielded critical insights into local $CO_2$ distribution. Notably, the highest $CO_2$ concentrations were recorded at Node 2, located near Pablo Zamora Avenue and La Rotonda Park. Despite its proximity to green spaces, persistently high vehicular traffic in this zone resulted in elevated emissions, suggesting that urban vegetation alone cannot offset pollution in high-traffic corridors, although these unusual values could also be caused by abnormal operation of the sensor. These findings provide a valuable empirical foundation for urban air quality management strategies.

This study delineates four pivotal directions for subsequent research aimed at advancing urban air quality monitoring systems: (1) the development of integrated multi-sensor networks capable of concurrently monitoring Particulate Matter sized 2.5 microns ($PM_{2.5}$) and $NO_2$ concentrations, facilitating more comprehensive pollution assessment; (2) investigation of the relationship between dynamic traffic flows and $CO_2$ dispersion patterns through real-time spatiotemporal analytics to enhance understanding of urban emission dynamics; (3) the deployment of machine learning algorithms to analyze historical sensor data for predicting pollution hotspots, enabling proactive environmental management; and (4) the evaluation of hybrid LoRaWAN-5G communication architectures to augment network reliability and scalability within high-density urban environments, addressing current limitations in data transmission efficiency. These proposed avenues of inquiry are anticipated to substantially contribute to the development of more robust, intelligent, and responsive air quality monitoring infrastructure.

**Author Contributions:** Conceptualization, K.E.S.-B., M.N. and J.L.Z.-M.; data curation, K.E.S.-B.; formal analysis, K.E.S.-B.; investigation, K.E.S.-B.; methodology, M.N. and J.L.Z.-M.; software, K.E.S.-B.; supervision, M.N. and J.L.Z.-M.; validation, M.N. and J.L.Z.-M.; visualization, K.E.S.-B., M.N. and J.L.Z.-M.; writing—original draft, K.E.S.-B.; writing—review and editing, M.N. and J.L.Z.-M. All authors have read and agreed to the published version of the manuscript .

**Funding:** This research received no external funding.

**Institutional Review Board Statement:** Not applicable.

**Informed Consent Statement:** Not applicable.

**Data Availability Statement:** The dataset of $CO_2$ measurements is available at https://github.com/mnaviam/CO2dataPVO (accessed on 11 September 2025).

**Acknowledgments:** The authors gratefully acknowledge the Universidad Técnica de Manabí for providing the institutional support, infrastructure, and resources necessary to conduct this research. We extend our sincere appreciation to the Postgraduate Faculty for fostering an environment conducive to scientific inquiry and innovation.

**Conflicts of Interest:** The authors declare no conflicts of interest.

## Abbreviations

The following abbreviations are used in this manuscript:

| | |
|---|---|
| $CO_2$ | Carbon Dioxide |
| CO | Carbon Monoxide |
| ESPE | University of the Armed Forces |
| FDS | Fire Dynamic Simulator |
| IoT | Internet of Things |
| LoRa | Long-Range |
| LoRaWAN | Long-Range Wide Area Network |
| LPWAN | Low-Power Wide Area Network |
| $PM_{2.5}$ | Particulate Matter sized 2.5 microns |
| NetSim | Network Simulator |
| $NO_2$ | Nitrogen Dioxide |
| MQTT | Message Queuing Telemetry Transport |
| UTM | Universidad Técnica de Manabí |
| WSN | Wireless Sensor Network |
| WSNs | Wireless Sensor Networks |

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
