# Peer review of "Optimizing CO2 Monitoring: Evaluating a Sensor Network Design"

_jsan, doi:10.3390/jsan14050093_

Round 1
Reviewer 1 Report
Comments and Suggestions for Authors
The manuscript presents the design and evaluation of a wireless sensor network (WSN) for COâ‚‚ monitoring in Portoviejo City, Ecuador. The authors combine simulation (via CupCarbon) and physical implementation (using LoRaWAN and MQ-135 sensors) to assess packet loss and network reliability. The topic is of interest, given the rising importance of urban air quality monitoring. The manuscript is generally well structured, with clear objectives, methodology, and results. However, there are several methodological, technical, and presentation issues that must be addressed before the manuscript can be considered for publication. The reviewer therefore recommends major revision at this stage.
Here are the detailed comments and concerns from the reviewer.
- Sensor accuracy and limitations: The MQ-135 sensor is a low-cost device with known cross-sensitivities to gases other than CO2. The manuscript does not sufficiently acknowledge these limitations or explain calibration procedures. Without calibration or validation against reference-grade instruments, the reliability of the reported COâ‚‚ data is questionable. Please provide details on calibration, accuracy, and possible error margins of the MQ-135 in the reported deployment.
- Statistical analysis: While Welch’s t-tests are reported, the choice of significance level (alpha=0.01) and the interpretation of results are inconsistent. For example, p=0.048 is described as supporting similarity, but under alpha=0.01 this is non-significant. The interpretation should be revised with consistent statistical reasoning. In addition, the small sample size (e.g., only 3 days of data in some cases) severely limits statistical power. Please clarify how robust these conclusions are.
- Environmental factors: The discussion attributes packet loss differences to distance, buildings, and weather, but these claims remain speculative without supporting measurements (e.g., signal strength, RSSI/SNR data, or weather logs, etc.). Providing quantitative evidence would strengthen the analysis.
- Novelty and contribution: Numerous prior studies have applied LoRaWAN and low-cost gas sensors for urban air monitoring. As currently framed, the novelty claim is limited. Please clarify whether the unique contribution lies in the Portoviejo case study, in the combined simulation–implementation methodology, or in another aspect. Highlighting this distinction more explicitly will strengthen the manuscript’s positioning.
- Data quality and interpretation: The CO2 concentration data presented in Figure 7 should be interpreted with caution. Reported ppm values exceed typical urban outdoor ranges (~400–600 ppm baseline, with peaks up to 800–1000 ppm in heavy traffic). Values above 1200–1300 ppm outdoors could be unusual and should either be justified (e.g., specific local conditions) or clearly noted as potential sensor limitations.
- Writing and presentation: While the English is understandable, the manuscript would benefit from language polishing to improve grammar, conciseness, and readability. Several sections (e.g., the Related Works) read as long lists of references and should be synthesized into a more critical comparison. Figures also require higher resolution and clearer legends for better readability.
Author Response
Title: Optimizing CO2 Monitoring: Evaluating a Sensor Network Design
Authors: Kenia Elizabet Sabando Bravo, Marlon Navia, Jorge Luis Zambrano-Martinez
Date: 2025-09-05
Journal: Journal of Sensor and Actuator Networks
Issue: Applications of Wireless Sensor Networks: Innovations and Future Trends
Manuscript Number: jsan-3820463
First, we would like to thank the reviewers for their constructive comments on the original version of this paper. We have incorporated all the changes suggested in the new version of the paper. We believe that the current version of the paper has adequately addressed all comments, along with their corresponding explanations.
We have prepared the details of the suggested changes, summarizing, in boldface, the issues pointed out by the reviewers. All the changes have been incorporated into the revised version of our original manuscript.
Reviewer # 1:
The manuscript presents the design and evaluation of a wireless sensor network (WSN) for COâ‚‚ monitoring in Portoviejo City, Ecuador. The authors combine simulation (via CupCarbon) and physical implementation (using LoRaWAN and MQ-135 sensors) to assess packet loss and network reliability. The topic is of interest, given the rising importance of urban air quality monitoring. The manuscript is generally well structured, with clear objectives, methodology, and results. However, there are several methodological, technical, and presentation issues that must be addressed before the manuscript can be considered for publication. The reviewer therefore recommends major revision at this stage.
Here are the detailed comments and concerns from the reviewer.
- Sensor accuracy and limitations: The MQ-135 sensor is a low-cost device with known cross-sensitivities to gases other than CO2. The manuscript does not sufficiently acknowledge these limitations or explain calibration procedures. Without calibration or validation against reference-grade instruments, the reliability of the reported COâ‚‚ data is questionable. Please provide details on calibration, accuracy, and possible error margins of the MQ-135 in the reported deployment.
Reply: Thank you for your comments and acceptance of the article. We have added an explanation of the sensor module calibration, including the equation used to determine CO2 ppm. A missing detail is the error range, because the manufacturer does not provide this data.
Even if the MQ-135 sensor is a low-cost component and therefore not too accurate, it can still be useful for evaluating the WSN.
- Statistical analysis: While Welch’s t-tests are reported, the choice of significance level (alpha=0.01) and the interpretation of results are inconsistent. For example, p=0.048 is described as supporting similarity, but under alpha=0.01 this is non-significant. The interpretation should be revised with consistent statistical reasoning. In addition, the small sample size (e.g., only 3 days of data in some cases) severely limits statistical power. Please clarify how robust these conclusions are.
Reply: Thank you for your comments and acceptance of the article. Although an alpha of 0.01 does not strictly imply robustness, an adequate value for significance can be relative, according to the type of study. However, we recognize that the analysis could not be too robust, as we affirmed in the first version of the manuscript. We have revised certain sentences in Section 4.3 to clarify these analyses and acknowledge their potential limitations.
- Environmental factors: The discussion attributes packet loss differences to distance, buildings, and weather, but these claims remain speculative without supporting measurements (e.g., signal strength, RSSI/SNR data, or weather logs, etc.). Providing quantitative evidence would strengthen the analysis.
Reply: Thank you for your comments and acceptance of the article. Indeed, on the first day of the test, it rained in Portoviejo city (historical data that confirms the occurrence is available https://www.tutiempo.net/clima/ws-841350.html). The RSSI was obtained from simulation but not from physical implementation.
- Novelty and contribution: Numerous prior studies have applied LoRaWAN and low-cost gas sensors for urban air monitoring. As currently framed, the novelty claim is limited. Please clarify whether the unique contribution lies in the Portoviejo case study, in the combined simulation–implementation methodology, or in another aspect. Highlighting this distinction more explicitly will strengthen the manuscript’s positioning.
Reply: Thank you for your comments and acceptance of the article. The manuscript's contributions include both the case study and the methodology applied, as well as the CO2 values obtained in the tests. These contributions are now remarked in the Introduction.
- Data quality and interpretation: The CO2 concentration data presented in Figure 7 should be interpreted with caution. Reported ppm values exceed typical urban outdoor ranges (~400–600 ppm baseline, with peaks up to 800–1000 ppm in heavy traffic). Values above 1200–1300 ppm outdoors could be unusual and should either be justified (e.g., specific local conditions) or clearly noted as potential sensor limitations.
Reply: Thank you for your comments and acceptance of the article. We have considered this comment and noted the possibility that an unusual operation or sensor limitation may be the cause of these values in Node 2.
- Writing and presentation: While the English is understandable, the manuscript would benefit from language polishing to improve grammar, conciseness, and readability. Several sections (e.g., the Related Works) read as long lists of references and should be synthesized into a more critical comparison. Figures also require higher resolution and clearer legends for better readability.
Reply: Thank you for your comments and acceptance of the article. We have attempted to improve the readability of the English language. Similarly, we have updated some figures that previously had small legends.

Reviewer 2 Report
Comments and Suggestions for Authors
1) The sentence in the abstract: “This methodology considers the development and evaluation of some 3 scenarios: i) Physical implementation scenario A, ii) Simulation scenario A, and iii) Simulation 4 scenario B, both by using the CupCarbon software.” Is too vague. Please rephrase and improve it.
2) Please restate the statement of contributions so as to better underline the technical challenges tackled.
3) Related work discussion should be complemented with a table categorizing the reviewed works along their main distinctive axes so as to better position the present work.
4) Related to the above comment, the literature discussion on sensor networks/systems for Co2 monitoring (in different applications) is somewhat scarce, see e.g.:
Tabella, Gianluca, et al. "Time-aware distributed sequential detection of gas dispersion via wireless sensor networks." IEEE Transactions on Signal and Information Processing over Networks 9 (2023): 721-735.
Bai, Xingzhen, et al. "Collaborative fusion estimation over wireless sensor networks for monitoring CO2 concentration in a greenhouse." Information Fusion 42 (2018): 119-126.
5) Sec. 3.2: I would like the authors to discuss and motivate in more detail the two displacement scenarios.
6) It is not clear to me whether the real data acquired from the Co2 sensors and whose data analysis has been carried out, are (or will be) made publicly available by the authors for reproducibility.
7) A slightly more detailed data analysis (other than mean, variance and percentiles) should be provided by the authors, e.g. data or temporal correlation among the measurements.
Author Response
Title: Optimizing CO2 Monitoring: Evaluating a Sensor Network Design
Authors: Kenia Elizabet Sabando Bravo, Marlon Navia, Jorge Luis Zambrano-Martinez
Date: 2025-09-05
Journal: Journal of Sensor and Actuator Networks
Issue: Applications of Wireless Sensor Networks: Innovations and Future Trends
Manuscript Number: jsan-3820463
First, we would like to thank the reviewers for their constructive comments on the original version of this paper. We have incorporated all the changes suggested in the new version of the paper. We believe that the current version of the paper has adequately addressed all comments, along with their corresponding explanations.
We have prepared the details of the suggested changes, summarizing, in boldface, the issues pointed out by the reviewers. All the changes have been incorporated into the revised version of our original manuscript.
Reviewer # 2:
- The sentence in the abstract: “This methodology considers the development and evaluation of some 3 scenarios: i) Physical implementation scenario A, ii) Simulation scenario A, and iii) Simulation 4 scenario B, both by using the CupCarbon software.” Is too vague. Please rephrase and improve it.
Reply: Thank you for your comments and acceptance of the article. We have rewritten the sentence to improve its clarity and ease of reading.
- Please restate the statement of contributions so as to better underline the technical challenges tackled.
Reply: Thank you for your comments and acceptance of the article. Now that the contributions have been noted in the Introduction section, they are detailed using bullets.
- Related work discussion should be complemented with a table categorizing the reviewed works along their main distinctive axes so as to better position the present work.
Reply: Thank you for your comments and acceptance of the article. Besides the addition of other references (see next question), we added a Table to categorize the cited related works. We consider three aspects: communication technology coverage, measurement mode (simple or data fusion), and the method for evaluating each proposal.
- Related to the above comment, the literature discussion on sensor networks/systems for Co2 monitoring (in different applications) is somewhat scarce, see e.g.:
Tabella, Gianluca, et al. "Time-aware distributed sequential detection of gas dispersion via wireless sensor networks." IEEE Transactions on Signal and Information Processing over Networks 9 (2023): 721-735.
Bai, Xingzhen, et al. "Collaborative fusion estimation over wireless sensor networks for monitoring CO2 concentration in a greenhouse." Information Fusion 42 (2018): 119-126.
Reply: Thank you for your comments and acceptance of the article. We have added the recommended references, as well as additional ones, to enhance the Related Works section.
- Sec. 3.2: I would like the authors to discuss and motivate in more detail the two displacement scenarios.
Reply: Thank you for your comments and acceptance of the article. We consider that Section 3.2 explains the motivation for the node's location in both scenarios. However, we added a paragraph to emphasize the selection of the locations.
- It is not clear to me whether the real data acquired from the CO2 sensors and whose data analysis has been carried out, are (or will be) made publicly available by the authors for reproducibility.
Reply: Thank you for your comments and acceptance of the article. Currently, the acquired data are available on open MQTT channels, from which they can be downloaded, and more data can also be added. To enable reproducibility, we will make the specific range of data considered in this study (CO2 monitored values) available as part of the paper.
- A slightly more detailed data analysis (other than mean, variance and percentiles) should be provided by the authors, e.g. data or temporal correlation among the measurements.
Reply: Thank you for your comments and acceptance of the article. We have added an analysis of the CO2 measurements obtained from the physical implementation. This analysis is in section 4.1.

Reviewer 3 Report
Comments and Suggestions for Authors
- The sensor network design for monitoring carbon dioxide (CO2) pollution in Portoviejo City, Ecuador, was evaluated through a methodology that combines simulation and physical implementation. In these scenarios, the functionality of Wireless Sensor Networks (WSN) is analyzed by implementing the LoRaWAN communication technology (Table 3). Furthermore, using the MQ-135 sensor, obtaining data on the PPM of (CO2) to examine the areas that concentrate the most significant amount of this atmospheric pollution was possible. Please elaborate the reasons using LoRaWAN communication technology in detail.
- What are the advantages and disadvantages of the CO2-based sensor network design compared to other designs? Please provide some insightful discussions of the CO2-based sensor network design to demonstrate your contribution in detail.
- In the figures 2 and 3, Scenario A and B designs should be demonstrated in detail.
- The original manuscript has no equations; the number of the equations should be increased to retain the reader engagement.
Author Response
Title: Optimizing CO2 Monitoring: Evaluating a Sensor Network Design
Authors: Kenia Elizabet Sabando Bravo, Marlon Navia, Jorge Luis Zambrano-Martinez
Date: 2025-09-05
Journal: Journal of Sensor and Actuator Networks
Issue: Applications of Wireless Sensor Networks: Innovations and Future Trends
Manuscript Number: jsan-3820463
First, we would like to thank the reviewers for their constructive comments on the original version of this paper. We have incorporated all the changes suggested in the new version of the paper. We believe that the current version of the paper has adequately addressed all comments, along with their corresponding explanations.
We have prepared the details of the suggested changes, summarizing, in boldface, the issues pointed out by the reviewers. All the changes have been incorporated into the revised version of our original manuscript.
Reviewer # 3:
- The sensor network design for monitoring carbon dioxide (CO2) pollution in Portoviejo City, Ecuador, was evaluated through a methodology that combines simulation and physical implementation. In these scenarios, the functionality of Wireless Sensor Networks (WSN) is analyzed by implementing the LoRaWAN communication technology (Table 3). Furthermore, using the MQ-135 sensor, obtaining data on the PPM of (CO2) to examine the areas that concentrate the most significant amount of this atmospheric pollution was possible. Please elaborate the reasons using LoRaWAN communication technology in detail.
Reply: Thank you for your comments and acceptance of the article. The primary reasons for utilizing LoRaWAN technology are its wide coverage, low energy consumption, and relatively low cost. Most WWAN proposals in related work use this technology for communication. This point is remarked in Section 3.1.1.
- What are the advantages and disadvantages of the CO2-based sensor network design compared to other designs? Please provide some insightful discussions of the CO2-based sensor network design to demonstrate your contribution in detail.
Reply: Thank you for your comments and acceptance of the article. The main contribution of this manuscript is not only the design of the WSN for CO2 monitoring, but also the proposed methodology and preliminary measurements in Portoviejo city, as mentioned in the Introduction. Additionally, we have not found another proposal for a WSN in the scientific literature for monitoring CO2 or other gases in this city; however, we have included some comments on this topic in the Discussion section.
- In the figures 2 and 3, Scenario A and B designs should be demonstrated in detail.
Reply: Thank you for your comments and acceptance of the article. We believe there is an error in this observation, as Figure 3 pertains to the sensor node, not a scenario (please refer to Figures 1 and 2). Mentioned figures (1 and 2) show the location of the nodes in each design. In the paragraphs that cite these images, the designs are detailed, as well as in Tables 2 and 3.
- The original manuscript has no equations; the number of the equations should be increased to retain the reader engagement.
Reply: Thank you for your comments and acceptance of the article. We have added equations in the manuscript to specify some calculations.

Round 2
Reviewer 1 Report
Comments and Suggestions for Authors
The reviewer would like to thank the authors for the reply and revision. The manuscript has been improved and it is suitable for publication.
Reviewer 2 Report
Comments and Suggestions for Authors
The authors have addressed all my previous comments.
Reviewer 3 Report
Comments and Suggestions for Authors
No further comment.